# Application of Kombucha Fermentation Broth for Antibacterial, Antioxidant, and Anti-Inflammatory Processes

**DOI:** 10.3390/ijms241813984

**Published:** 2023-09-12

**Authors:** Jingqian Su, Qingqing Tan, Shun Wu, Bilal Abbas, Minhe Yang

**Affiliations:** 1Fujian Key Laboratory of Innate Immune Biology, Biomedical Research Center of South China, College of Life Science, Fujian Normal University, Fuzhou 350117, China; tqq4507@163.com (Q.T.); qsx20221412@student.fjnu.edu.cn (S.W.); bilalabbas857@gmail.com (B.A.); 2Fujian Key Laboratory of Microbial Pathogenesis and Interventions-Fujian Province University, Biomedical Research Center of South China, College of Life Science, Fujian Normal University, Fuzhou 350117, China

**Keywords:** kombucha, turmeric, *Paeoniae alba*, antibacterial, anti-inflammatory

## Abstract

Treatment for sepsis and its complications in the clinic is primarily in the forms of antibiotics, anti-inflammatory agents, and antioxidant drugs. Kombucha, a traditional fermented beverage rich in tea polyphenols and organic acids, offers several benefits including bacteriostasis, anti-inflammation ability, and boosting the immune system. Currently, research on kombucha is primarily focused on its antibacterial and antioxidant properties; however, in-depth exploration of the involved mechanisms is lacking. Herein, turmeric, *Paeoniae alba*, and black tea were used as fermentation substrates to detect the bacteriostatic and antioxidant activities of the fermentation broth and evaluate its anti-inflammatory effects on RAW264.7 cells stimulated by lipopolysaccharides (LPSs). The results showed that fermentation enhanced the antibacterial activity of turmeric against *E. coli* and *S. aureus* and that of *Paeoniae alba* against *S. aureus*. Turmeric black tea exhibited the highest antioxidant activity. The fermentation broth of turmeric and turmeric black tea significantly reduced the expression of inflammatory cytokines induced by LPSs. Our results showed that using turmeric and *Paeoniae alba* culture media as substrates can enhance the anti-inflammatory effects of fermentation broth and provide a new strategy for developing anti-inflammatory substances.

## 1. Introduction

The latest definition of sepsis is life-threatening organ dysfunction caused by a dysregulated host response to infection [1]. It remains a prominent global health issue, affecting approximately 18 million people, with mortality rates ranging from 28% to 50%. Currently, no specific pharmacological interventions have been specifically designed for sepsis treatment [2]. The primary therapeutic strategy involves managing excessive inflammation during infection [3]. In clinical settings, the management and treatment of sepsis and its associated complications primarily involve the administration of antibiotics, anti-inflammatory agents, and antioxidant medications [4].

Kombucha is an acidic tea beverage fermented using acetic acid bacteria, yeast, and lactic acid bacteria, which are rich in tea polyphenols and organic acids and have antioxidant, anti-inflammatory, and enhanced immunity properties [5,6]. It can be prepared from tea and other raw materials. Considering the beneficial role of kombucha, studies on alternative substrates such as coconut water have emerged [7].

Owing to antibiotics’ abuse and the emergence of multidrug-resistant strains, traditional Chinese medicine has received increasing attention from researchers [8]. Microbial fermentation is an important process in traditional Chinese medicine that plays a pharmacological role. The high acidity of kombucha provides a new strategy for controlling pathogens and food spoilage bacteria [9]. In human diseases related to overexpression of reactive oxygen species (ROS) and free radicals, including inflammatory diseases, cancer, senile diabetes, neurodegenerative diseases, and arteriosclerosis, antioxidants and enzymes can be used to scavenge these species [10]. The rich phenolic content and flora types in kombucha cultures provide the natural advantage of antioxidant activity [11].

Traditional kombucha has anti-inflammatory effects, and certain studies have shown that broths that are fermented with alternative materials have anti-inflammatory properties. Wang et al. found that kombucha can improve cellular immune function disorders in the early stages of sepsis in mice, promote the growth of butyric-acid-producing bacteria, and exert anti-inflammatory effects [12]. Cabral et al. determined that oak leaves fermented by kombucha can effectively reduce NO production and down-regulate the expression levels of interleukin (IL)-1β and tumor necrosis factor-alpha (TNF-α) in macrophages [13]. Ziemlewska et al. found that Madai tea fermented with kombucha showed strong anti-LOX activity and had the potential to treat inflammation [14]. Few studies have reported on the use and efficacy of traditional Chinese medicines, such as Radix *Paeoniae alba* and turmeric, as substrates. Radix *P. alba* is an herbal medicine with anti-inflammatory and immunomodulatory activities. Its primary compounds are flavonoids, polyphenols, and monoterpene glycosides [15]. 

Turmeric is not only a natural food seasoning and pigment material but also a traditional Chinese medicine containing bioactive compounds, among which polyphenols and curcumin derivatives possess antioxidant, anti-inflammatory, and anticancer properties [16]. Like black tea, both are rich in polyphenols and have the potential to be used as substitutes for black tea. Turmeric and Radix *P. alba* also have anti-inflammatory effects [17]. The lipopolysaccharide (LPS) stimulation of RAW264.7 can activate a series of signal pathways, such as nuclear factor kappa B (NF-κB) and mitogen-activated protein kinase (MAPK), resulting in numerous inflammatory cytokine expressions, which is a good model for in vitro inflammation research [18,19]. But it can only simulate inflammation and cannot simulate infection in sepsis, etc., so it is not realistic enough.

However, the quest for a pharmaceutical intervention possessing combined antibiotic, anti-inflammatory, and antioxidant properties specifically tailored for sepsis treatment remains ongoing. Our previous studies have demonstrated that kombucha can act as an anti-inflammatory agent and prevent the development of systemic inflammatory responses associated with sepsis [12]. Therefore, in this study, turmeric, *Paeoniae alba*, and black tea were used as substrates to prepare a fermentation broth to compare the bacteriostatic rate, antioxidation ability, and the effect of different substrates on the expression of inflammatory factors (IL-6, IL-1β, TNF-α, CXCL10, etc.). The results of this study are expected to indicate that kombucha can be used as a potent anti-inflammatory agent for sepsis prevention and treatment. In addition, we need more experiments to prove the mechanism of turmeric kombucha in the treatment of sepsis and its feasibility in the clinic.

## 2. Results

### 2.1. Bacteriostatic Activity of Tea Culture Broth on E. coli and S. aureus

The bacteriostatic rate of the tea fermentation liquid increased to 100% with increasing concentrations (Figure 1a–h). The tumor medium had an antibacterial effect on *E. coli* after only 14 d, with the highest attaining 62.35%, which was lower than that of the fermentation broth of the turmeric kombucha (*p* < 0.0001; Figure 1b), and it had no inhibitory effect on *S. aureus* (Figure 1c,d). The maximum antibacterial activity of the extract of *Paeoniae alba* culture medium (PW) root against *E. coli* increased from 84.76% (7 d) to 99.97% (14 d) with time (Figure 1e,f). At 7 and 14 d, no significant difference was observed in the antibacterial effect of the *Paeoniae alba* extract on *S. aureus*, and the activity of *S. aureus* could be 100% inhibited with the increase of the concentration factor (Figure 1g,h). After 14 d, the antibacterial activity of the three fermentation broths against *S. aureus* was increased to the same level as that of the Radix PW extract (Figure 1h; Table 1 and Table 2). The results show that fermentation enhanced the antibacterial activity of the turmeric kombucha but did not significantly increase the antibacterial activity against *S. aureus* of the PW (Figure 1a–h; Table 1, Table 2, Table 3 and Table 4). Moreover, the PW kombucha had the strongest antibacterial effect on *E. coli*, and the mixed fermentation substrates had a synergistically enhanced antibacterial effect, while *Paeoniae alba* extract had the strongest antibacterial effect on *S. aureus.*

### 2.2. Antioxidant Activity of Kombucha Fermentation Broth

Fermentation did not significantly improve the antioxidant activity of turmeric (Figure 2b). The DPPH free radical scavenging ability of turmeric kombucha fermentation liquid was higher than that of the kombucha and turmeric fermentation liquids, and the DPPH scavenging rate of *C. kombucha* and kombucha fermentation liquids was higher than that of vitamin C (0.5 mg/mL; Figure 2a–c). The antioxidant capacity of the PK was not significantly improved compared to that of that PW (*p* > 0.05); however, the antioxidant capacity of the kombucha fermented liquid was superior. The scavenging abilities of the fermentation liquids of the PW and kombucha to DPPH were lower than that of the kombucha and PW (Figure 2f,g). When the concentration factor was 1.25×, only kombucha could eliminate DPPH within 30 min. The DPPH scavenging rates of the PW, *Kombucha alba*, and PW extracts showed a downward trend (Figure 2d,e,h,i). The antioxidant activity of the turmeric kombucha (TK) fermented liquid was the highest, and the antioxidant activities of the fermented and extract liquids gradually increased with an increase in the concentration ratio. Moreover, the antioxidant activity tended to flatten after attaining a certain concentration except for in *Paeoniae alba* kombucha, where the antioxidant activity of the fermentation broth decreased slightly with an increase in the concentration ratio (Figure 2a–h).

### 2.3. Cytotoxicity of Kombucha Fermentation Broth to L929 and RAW264.7 Cells

Prolonging the fermentation time increased the toxicity of the TK and turmeric kombucha to L929 cells, while prolonging the standing time reduced the inhibitory effect of the turmeric medium (TW) on the viability of L929 cells (Figure 3a,b). The turmeric kombucha and TK had no effect on the viability of RAW264.7 cells (Figure 3c,d). The PW kombucha at concentrations of 10 and 500 μg/mL could promote the proliferation of L929 cells (*p* < 0.05). The prolongation of fermentation time reduced the growth rate of the PW and TK (Figure 3e,f). TPK could significantly promote the proliferation of RAW264.7 cells, but the prolongation of fermentation time reduced the promoting effect. When the concentration of the PW was greater than 100 g/mL, it had obvious cytotoxicity to RAW264.7 cells (Figure 3g,h). The fermentation broth and lyophilized powder from the turmeric group had no inhibitory effect on the viability of RAW264.7 cells stimulated by LPS (Figure 4a–p).

### 2.4. Effects of Lyophilized Powder in Turmeric Groups on the Expression Level of Cellular Inflammatory Factors Caused by LPS

Kombucha, turmeric, turmeric fermentation broth, and TW alone did not cause an inflammatory response in RAW264.7 cells (Figure 5 and Figure 6). Turmeric kombucha, turmeric fermentation broth, and TW (7 d) fermented for 7 d had a significant inhibitory effect on LPS-induced inflammatory factors (i.e., TNF-α, IL-1β, and CXCL10; Figure 5j–l,r–t; Figure 6b–d), and turmeric kombucha fermentation broth (14 d) could significantly reduce IL-6, IL-1β, and CXCL10 expression levels (Figure 5g,w and Figure 6g). TW significantly reduced CXCL10 expression levels (Figure 6b,f). Simultaneously, kombucha fermentation broth and turmeric kombucha fermentation broth (14 d) had no anti-inflammatory effect (Figure 5a,e,h,i,m,p,q,u,x; Figure 6a,e,h,i,m,p), and kombucha, turmeric kombucha, turmeric fermentation broth, and TW had no inhibitory effect on the LPS-induced increase in NLRP3 expression (Figure 6i–p). In addition, TW increased the expression level of IL-6 in LPS-induced RAW264.7 cells (Figure 5b,f).

## 3. Discussion

Turmeric extract itself does not completely inhibit the growth of *E. coli* and *S. aureus,* but they are inhibited by fermented curcumin. The inhibitory abilities against *S. aureus* and *E. coli* were significantly improved by treatment with fermented curcumin. Kombucha fermentation enhanced the inhibitory effect of *P. alba* on *E. coli*. Prolonged fermentation time increased the inhibitory activity of the turmeric extract against *E. coli* while enhancing the inhibitory effects of kombucha and *C. longa* on *S. aureus*. Prolonging the fermentation time improved the inhibition of *E. coli* by samples from the *P. alba* group and enhanced the inhibitory effect of the fermented *P. alba* liquid on *S. aureus*. Among the three different fermentation substrates in the turmeric group, traditional black tea showed stronger antibacterial effects against *E. coli* and *S. aureus* compared to that for turmeric as the substrate. Similarly, of the three different fermentation substrates of the *P. alba* group, *P. alba* exhibited stronger antibacterial effects against *E. coli* and *S. aureus* than black tea, and the mixed substrates demonstrated synergistic enhancement in inhibiting *E. coli.* Traditional kombucha exhibited higher antioxidant capacity than domesticated *C. longa* and *Paeoniae alba*.

Low-concentration kombucha fermentation broth promoted the growth of the two bacteria, presumably because of the low content of active substances at low concentrations, sufficient medium, and original carbohydrates in the fermentation broth, thereby providing an excellent growth environment for the bacteria. In this study, the sugar content of the turmeric group was higher than that of the *Paeoniae alba* group (35.0 g/L vs. 30.0 g/L), and the negative value in the turmeric group was larger than that in the *Paeoniae alba* group. This result confirms the previous speculation.

Among the bioactive compounds found in kombucha fungi, substances that play an antioxidant role are primarily organic acids and polyphenols. Phenolic compounds are primarily responsible for the antioxidant properties of kombucha because they can reduce free radicals and ROS, stimulate the activity of antioxidant enzymes, and inhibit pro-oxidative enzymes [20,21]. Phenolic compounds readily donate hydroxyl hydrogen owing to resonance stabilization [21,22], and this hydrogen supply endows phenolic compounds with an excellent DPPH scavenging ability [22,23]. Fermentation did not increase the antioxidant activity of turmeric and *Paeoniae alba*, probably because the polyphenols in turmeric and *Paeoniae alba* were transformed into substances without DPPH free radical scavenging activity during the fermentation process. Turmeric polysaccharides, curcumin, demethoxycurcumin, and bisdemethoxycurcumin act as scavengers of DPPH free radicals [23,24,25]. Phenolic acids, gallic acid tannins, paeoniflorin, benzoylpaeoniflorin, paeonol, paeonifloride C, benzoic acid, and other compounds in *P. alba* exhibit antioxidant activity [15,25,26]. Fermentation with different strains affects the antioxidant capacities of turmeric and *Paeoniae alba*.

Cheng et al. showed that at a concentration of 31.25 μg/mL, the DPPH clearance rate of *Lactobacillus*-fermented turmeric (7.88 ± 3.36%) was lower than that of unfermented turmeric (9.36 ± 4.24%) [26,27], which is similar to our experimental results. However, similar studies have used different strains that show enhanced antioxidant activity in turmeric/*Paeoniae alba* after fermentation. Plangpin et al. isolated and screened *Lactobacillus plantarum* from turmeric rhizomes as a starter for turmeric beverages and used DPPH free radical scavenging activity and FRAP assays to evaluate the antioxidant activity of fermented turmeric.

The fermentation process resulted in increased antioxidant activity [20,27]. Juho et al. found that compared with unfermented turmeric, *Rhizopus oligosporus*-fermented turmeric had higher concentrations of curcumin, demethoxycurcumin, bisdemethoxycurcumin, phenolic compounds, and total flavonoids. After 3 d of fermentation, the oxygen radical absorbance capacity and ferric reducing antioxidant power of turmeric increased by 1.47 and 2.25 times, respectively [26,27]. Shrijana et al. fermented *Paeoniae alba* root extract with plant-derived *Lactobacillus brevis* 174A, and the fermented *Paeoniae alba* root extract significantly increased the total phenolic content, decreased reactive oxygen species (ROS) levels in RAW264.7, and inhibited NO release [25,26]. The antibacterial activity of the turmeric fermentation liquid was higher than that of the turmeric extract. We speculate that the fermentation process was the primary reason for the antibacterial activity of turmeric, which was also the main reason for the enhanced activity of the *P. alba* fermentation liquid against *E. coli*.

The organic acids (acetic, gluconic, and glucuronic acids), tea polyphenols, and ethanol in traditional kombucha can inhibit bacteria and fungi in the fermentation broth. Acetic acid is the primary antimicrobial agent in kombucha, and the bacteriostatic activities of acetic and other organic acids are multifactorial. This includes the ability of undissociated acids to diffuse freely over the lipid bilayer and release protons into the cytoplasm, thereby lowering the pH. Undissociated acids at lower pH values become embedded in the lipid bilayers, and anions accumulate. The two main mechanisms affecting antimicrobial activity are cytoplasmic acidification and the accumulation of free acid anions at toxic levels [28,29]. Acidification of the bacterial cytoplasm can prevent growth by inhibiting glycolysis, preventing active transport, or interfering with signal transduction [29,30]. In addition to organic acids, other components introduced by alternative raw materials cannot be ignored owing to their antibacterial effects. Zubaidah et al. found that the antibacterial activity of turmeric kombucha increased with increasing turmeric concentration and decreased after reaching a certain level [30]. In this study, the extract of Radix *P. alba* exhibited a higher ability to inhibit *S. aureus* than the fermentation broth, which may be related to the unique substances present in Radix *P. alba*. Studies have shown that high concentrations of tellimagrandin I and pentagalloyl glucose in Radix *P. alba* have antibacterial activity, and telllimagrandin I can reduce antibiotic resistance and enhance the antibacterial effect of the antibiotics used [15].

In addition, we found that the kombucha fermentation liquid affected the activity of different cells. Following fermentation, the toxicity of turmeric and *Paeoniae alba* culture fluid to the cells was reduced. After LPS stimulated RAW264.7 cells, the inflammatory factors (IL-6, IL-1β, TNF-α, and CXCL10) significantly increased and were significantly inhibited after the interventions of *C. longa* and *C. kombucha* fermentation broth.

In L929 cells, both kombucha fermentation broth and medium have the potential to influence cell activity. The activity in L929 cells yielded distinct results for TW (14 d) and PW (7 d) media. Below 1000 μg/mL, as the concentration increased, cell activity exhibited a downward trend, whereas above 1000 μg/mL, with increasing concentration, cell activity displayed an upward trend. It is possible that above 1000 μg/mL, the cells develop a tolerance, or the physical or chemical properties of the culture are altered at higher concentrations, resulting in reduced cytotoxicity in the corresponding culture. Prolonging the fermentation time increased the cytotoxicity of the fermented liquid from the turmeric group on L929 cells but decreased the cytotoxicity of the fermented liquid from the *Paeoniae alba* group on L929 cells, yielding opposing results for the two groups. This phenomenon may be linked to the ratio of added tea sugar because the only difference between the preparations of kombucha in the two groups was the tea:sugar ratio. As the innate immune cells of mice, RAW264.7 appeared to be less affected by tea fermentation broth and culture medium than that for L929 cells. An increase in the concentration of *C. longa* or *Paeoniae alba* medium resulted in the inhibition of RAW264.7 cell activity.

Similarly, as the corresponding substrate, kombucha fermentation broth at concentrations up to 1000 μg/mL exhibited a proliferative effect on RAW264.7 cells as the concentration increased. Compared with the original substrate medium, the fermented substrate medium showed significantly reduced toxicity to RAW264.7 cells. At a concentration of 1000 μg/mL, whether fermented for 7 or 14 d, the turmeric fermentation broth exhibited superior proliferative activity on RAW264.7 cells compared to that for *P. alba* and turmeric–black tea fermentation liquids. Compared with traditional kombucha, substituting the fermentation substrate with turmeric resulted in greater cell activity at higher concentrations.

Upon LPS stimulation of RAW264.7 cells, the mRNA expressions of inflammatory factors, such as IL-6, IL-1β, TNF-α, and CXCL10, significantly increased (*p* < 0.05). After the intervention, the mRNA levels of the inflammatory factors were significantly reduced. Thus, TK and TBK may contain active components that inhibit the expression of inflammatory factors. Regarding mRNA transcription levels of IL-6, TNF-α, and IL-1β, the 7 d fermentation time was more effective than the 14 d period. Fermented *C. longa* was more effective at encouraging mRNA expressions of inflammatory factors than the unfermented turmeric culture solution, and the turmeric substrate performed better than black tea fermentation broth. For the CXCL10 mRNA levels, unfermented turmeric broth outperformed fermented tea culture broth. Studies have demonstrated that different strains of turmeric-fermented liquids exhibit similar anti-inflammatory effects. Turmeric fermented by lactic acid bacteria can inhibit the expression of TNF-α and Toll-like receptor 4 in LPS-stimulated RAW264.7 cells by inhibiting the c-Jun N-terminal kinase signaling pathway, thereby displaying anti-inflammatory activity.

In contrast, unfermented turmeric lacks this effect. Bayazid et al. found that *Clostridium butyricum*-fermented turmeric can reduce LPS-induced NO production in RAW264.7, revealing its anti-inflammatory effect [31]. Lim et al. demonstrated that turmeric fermented by *R. oligosporus* exhibited a 1.44-fold inhibitory effect on LPS-stimulated NO production by RAW264.7 cells after 3 d of fermentation (compared to unfermented turmeric) [27].

In summary, black tea kombucha-fermented liquids prepared from various substrates displayed excellent antibacterial properties. The fermentation process harnessed the antibacterial properties of both the raw materials and strains. Fermentation creates an acidic environment for the drug and confers antibacterial properties to raw materials. The active substances further contribute to bacteriostasis. The high acidity of the *Paeoniae alba* root and turmeric kombucha fermentation broth, along with the abundant polyphenols in the raw materials, endowed the fermentation broth with the ability to control pathogens and act as an antioxidant. Turmeric kombucha and turmeric kombucha-medicated serum substantially inhibited the LPS-induced increase in inflammatory factors, exhibiting anti-inflammatory activity. Compared to the TW, the turmeric fermentation liquid displayed a stronger protective effect and a more pronounced regulatory impact on the immune system [31]. Turmeric-based kombucha infusions have the potential to enhance the functional and nutritional values of kombucha. This study presents a new strategy for treating diseases caused by bacterial and ROS. These experimental data provide valuable insights for developing traditional Chinese medicine and new antibacterial and antioxidant drugs.

## 4. Materials and Methods

### 4.1. Chemicals and Reagents

Turmeric (yellow, fluffy powder) was provided by Mr. Fu Jianwei of the Fujian Academy of Agricultural Sciences, and *Paeoniae alba* (Bozhou, off-white powder) was provided by Professor Lu Yudong of Fujian Normal University. The remaining drugs and reagents are listed in Table 5.

### 4.2. Strains

(1) Kombucha: The kombucha mother liquor was obtained from Zhangzhou City, Fujian Province, and donated by Dr. Chen Zhihong from Fujian Normal University. (2) *E. coli* (ATCC25922) and *S. aureus* (ATCC6538) were donated by Dr. Wu Lina of Fujian Medical University.

### 4.3. Domestication and Culture of Black Tea Bacteria in Different Media Preparations of Fermentation Broth

Black tea bacteria were subcultured in TW, turmeric black tea, *Paeoniae alba* bacteria, and *Paeoniae alba* black tea media for 7 d until the bacteria grew stably, and the mother liquids of turmeric, turmeric black tea, *Paeoniae alba*, and *Paeoniae alba* black tea bacteria were obtained.

### 4.4. Preparation of Fermentation Broth

(1) Turmeric fermentation broth/L: 15.0 g turmeric powder, 35.0 g sugar, and 15% turmeric-tea-fermented mother liquid. (2) Fermentation broth/L of turmeric black tea: 15.0 g turmeric powder, 5.0 g of black tea, 35.0 g of granulated sugar, and 15% of turmeric–black-tea-fermented mother liquor. (3) Turmeric medium/L: 15 g turmeric powder. (4) Black tea bacteria/L: 5 g black tea, 35 g white granulated sugar, 15% black tea bacteria mother liquid *Paeoniae alba* group; the preparation method of fermentation broth in the *Paeoniae alba* group is similar, but the difference is as follows: 6 g black tea, 30 g white granulated sugar, 20 g *Paeoniae alba*.

### 4.5. Preparation of Freeze-Dried Powder Samples

The collected liquid samples were centrifuged at 5369× *g* for 10 min, and the supernatant (50 mL) was lyophilized for 48 h. The freeze-dried fermentation liquid powder was diluted with quantitative aseptic water and centrifuged at 7104× *g* for 5 min. The supernatant was filtered through a 0.22 μm microporous membrane.

### 4.6. Bacterial Culture and Preparation of Bacterial Liquid

*E. coli* and *S. aureus* were first inoculated on a Luria–Bertani (LB) solid medium plate via plate marking method and cultured in a biochemical incubator (37 °C) for 24 h to obtain a single colony. A bacterial suspension (1.5 × 10^6^ colony-forming unit (CFU) /mL) was prepared for follow-up experiments.

### 4.7. Antibacterial Activity Test (MIC)

LB liquid medium (150 μL) was spread onto a 96-well plate, and 150 μL freeze-dried concentrated fermentation broth was added in the first column. Then, 150 μL LB liquid medium was used as the solvent to repeat the double-dilution method, eight concentrations of fermentation liquid freeze-dried powder samples were prepared, 150 μL mixture in the eighth concentration sample well was discarded, and a 1.5 μL concentration of 1.5 × 10^6^ CFU/mL bacterial suspension (A1) was added to each of the first three wells. Sterilized saline (15 μL) was added to each row of the last three wells as the blank control (A2) of each concentration sample. Another row of the first three wells contained 15 μL bacterial suspension (without adding samples) as the bacterial growth control well (A0), and 15 μL sterilized saline was added into the latter three wells as the blank control (A3). After the 96-well plate was incubated in a biochemical incubator at 37 °C for 24 h, the MIC was determined by visual inspection under parallel light. The sample concentration was expressed as the freeze-dried concentration ratio of the fermentation broth (mL/mL).

### 4.8. Bacteriostatic Rate Determination

The OD value of 96-well plates at 600 nm was measured using an enzyme meter. The bacteriostatic rate was calculated according to the bacteriostatic rate formula, and the bacteriostatic curves of different bacteria under the action of freeze-dried concentrated samples with different concentrations were drawn. The inhibition rate (%) was determined as follows:Inhibition rate = [1 − (A1 − A2)/(A0 − A3)] × 100%(1)
where A1 is the absorbance of the medium containing fermentation broth and bacteria, A2 is the absorbance of the culture medium containing fermentation broth only, A0 is the absorbance of the culture medium containing bacteria only, and A3 is the absorbance of the culture medium alone.

### 4.9. Antioxidation Experiment

DPPH scavenging method was employed to detect the antioxidant activity [32].

### 4.10. Cell Line

RAW264.7 (mouse mononuclear macrophages) and L929 cells (mouse fibroblasts) were purchased from the American Type Culture ATCC (model culture collection).

### 4.11. CCK-8 Detection

Cell proliferation was evaluated using cell counting kit-8 (CCK-8). The density of the 96-well plates was 1 × 10^4^ cells/well (1:10). After incubation in a dark room at 37 °C for 1.5 h, the absorbance at 450 nm was measured, and the cell viability of each group was calculated according to the absorbance.

### 4.12. Cell Grouping Treatment

The six-well plates were loaded with approximately 1 × 10^5^ cells per well. The anti-inflammatory activity of the lyophilized powder was divided into four groups: blank (cell + medium + PBS), drug (drug + cell + medium), LPS (cell + medium + LPS), and drug + LPS (drug + cell + medium + LPS). The corresponding drug was then added to each sample (BK: 1 μg/mL; TW: 100 μg/mL; TK-7 d: 100 μg/mL; TK-14 d: 1000 μg/mL; TBK: 100 μg/mL). The medium was discarded and replaced with new medium, and LPS (1 μg/mL) was added for stimulation for 3 h. Then, the medium was removed, and Trizol was added to promote cleavage. The cells were then collected for further analysis.

### 4.13. Real-Time Quantitative PCR

qPCR was performed as previously described [12]. Primers are as follows: Glyceraldehyde-3-phosphate dehydrogenase (GAPDH)-F(5′ -CCGAGCTGAACGGGAAGCTCAC-3′), GAPDH-R(5′ -CCATGTAGGCCATGAGGTCCACC-3′); IL-6-F(5′ -TACCACTTCACAAGTCGGAGGC-3), IL-6-R(5′-RCTGCAAGTGCATCATCGTTGTTC-3′); IL-1β-F(5′-TGGGAAACAACAGTGGTCAGG-3′), IL-1β-R(5′-CCATCAGAGGCAAGGAGGAA-3′); TNF-α-F(5′-GAGTGACAAGCCTGTAGCC-3′), TNF-α-R(5′-CTCCTGGTATGAGATAGCAAA-3′); Chemokine(C-X-C motif) lig and 10 (CXCL10)-F(5′-CCAAGTGCTGCCGTCATT-3′); CXCL10-R(5′-GCTCATCATTCTTTTTCATCGT-3′); NLR family, pyrin domain-containing protein 3(NLRP3)-F(5′ -CAACAGTCGCTACACGCAG-3′), NLRP3-R(5′ -GTCCTCGGGCTCAAACAG-3′).

### 4.14. Statistical Analysis

All experimental data are expressed as x ± SD and were analyzed using the SPSS25.0 software package. A single-factor ANOVA was used to compare groups, and the least significant difference (LSD) test was used for pairwise comparison. The *t*-test was used to determine the significant difference between groups. A two-way ANOVA was used to compare the two factors. These differences were statistically significant (*p* < 0.05). ns: *p* > 0.05, *: *p* < 0.05, **: *p* < 0.01, ***: *p* < 0.001, ****: *p* < 0.0001.

## 5. Conclusions

Kombucha exhibits potent antibacterial properties and demonstrates antioxidant activity. Following fermentation, the cytotoxicities of turmeric and PW towards cells diminished, while the anti-inflammatory efficacy significantly amplified. These findings suggest that kombucha facilitates the biotransformation of turmeric constituents, thereby promoting antibacterial and anti-inflammatory effects. Using the LPS-induced cell inflammation model and established NF-κB inflammatory pathway as a starting point, this study provides a basis for the potential utilization of kombucha in the prevention and treatment of sepsis. However, the purification and action mechanism of the active substances in the kombucha broth require further investigations.

## Figures and Tables

**Figure 1 ijms-24-13984-f001:**
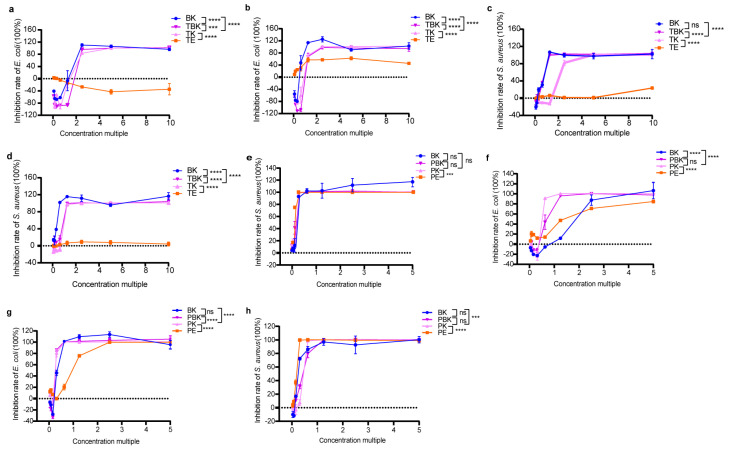
Changes in the inhibition rates of the freeze-dried powder samples in fermentation against *E. coli* and *S. aureus*. (**a**) Inhibition of *E. coli* by the lyophilized turmeric group (7 d); (**b**) inhibition of *E. coli* by the lyophilized turmeric group (14 d); (**c**) inhibition of *S. aureus* by the lyophilized turmeric group (7 d); (**d**) inhibition of *S. aureus* by the lyophilized turmeric group (14 d); (**e**) inhibition of *E. coli* by the lyophilized *Paeoniae alba* group (7 d); (**f**) inhibition of *E. coli* in the *Paeoniae alba* group (14 d); (**g**) inhibition of lyophilized powder on *S. aureus* in the *Paeoniae alba* group (7 d); (**h**) inhibition of lyophilized powder on *S. aureus* in the *Paeoniae alba* group (14 d). PE, *Paeoniae alba* extract; PK, *Paeoniae alba* kombucha; TPK, *Paeoniae alba*–black tea kombucha. Two-way analysis of variance (ANOVA) was used to compare the two factors. Statistical significance was set at *p* < 0.05. ns: *p* > 0.05; ***, *p* < 0.001; ****, *p* < 0.0001.

**Figure 2 ijms-24-13984-f002:**
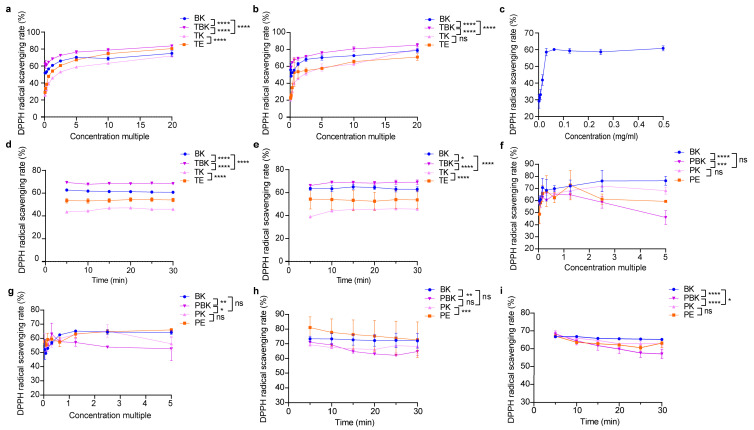
Lyophilized powder in the fermentation broth scavenges DPPH free radicals. (**a**) Scavenging of DPPH radicals by lyophilized powder (7 d) in the turmeric group; (**b**) scavenging of DPPH radicals by lyophilized powder (14 d) in the turmeric group; (**c**) scavenging of DPPH radicals by vitamin C; (**d**) 1.25-fold change in DPPH clearance of the turmeric group concentrate (7 d) with time; (**e**) 1.25-fold change in DPPH clearance of the turmeric group concentrate (14 d) with time; (**f**) scavenging of DPPH radicals by lyophilized powder (7 d) in the *Paeoniae alba* group; (**g**) Scavenging of DPPH radicals by lyophilized powder (14 d) in the *Paeoniae alba* group; (**h**) 1.25-fold change in DPPH clearance of the *Paeoniae alba* group concentrate (7 d) with time; (**i**) 1.25-fold change in DPPH clearance of the *Paeoniae alba* group concentrate (14 d) with time. PE, *Paeoniae alba* extract; PK, *Paeoniae alba* kombucha; TPK, *Paeoniae alba*–black tea kombucha. Two-way ANOVA was used to compare the two factors. Statistical significance was set at *p* < 0.05. ns: *p* > 0.05; *, *p* < 0.05; **, *p* < 0.01; ***, *p* < 0.001; ****, *p* < 0.0001.

**Figure 3 ijms-24-13984-f003:**
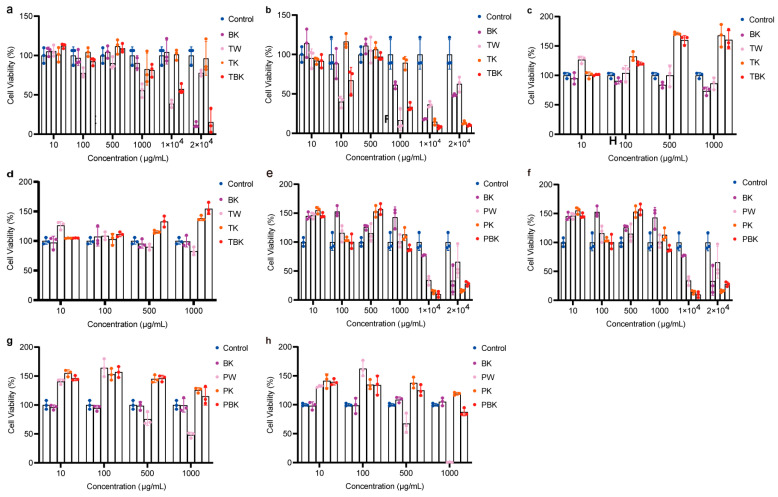
Different concentrations of the turmeric and *Paeoniae alba* fermentation broth-treated L929 and RAW264.7 cell viabilities following 24 h. (**a**) Effect of turmeric fermentation broth (7 d) on L929 cell viability; (**b**) effect of turmeric fermentation broth (14 d) on L929 cell viability; (**c**) effect of *Paeoniae alba* group fermentation broth (7 d) on L929 cell viability; (**d**) effect of *Paeoniae alba* group fermentation broth (14 d) on L929 cell viability; (**e**) effect of turmeric fermentation broth (7 d) on RAW264.7 cell viability; (**f**) effect of turmeric fermentation broth (14 d) on RAW264.7 cell viability; (**g**) effect of *Paeoniae alba* fermentation broth (7 d) on RAW264.7 cell viability; (**h**) effect of *Paeoniae alba* fermentation broth (14 d) on RAW264.7 cell viability. BK, kombucha; TW, turmeric medium; TK, turmeric kombucha; TBK, turmeric kombucha; PW, *Paeoniae alba* culture medium; PK, *Paeonia alba*; PBK, red *C. Paeoniae*.

**Figure 4 ijms-24-13984-f004:**
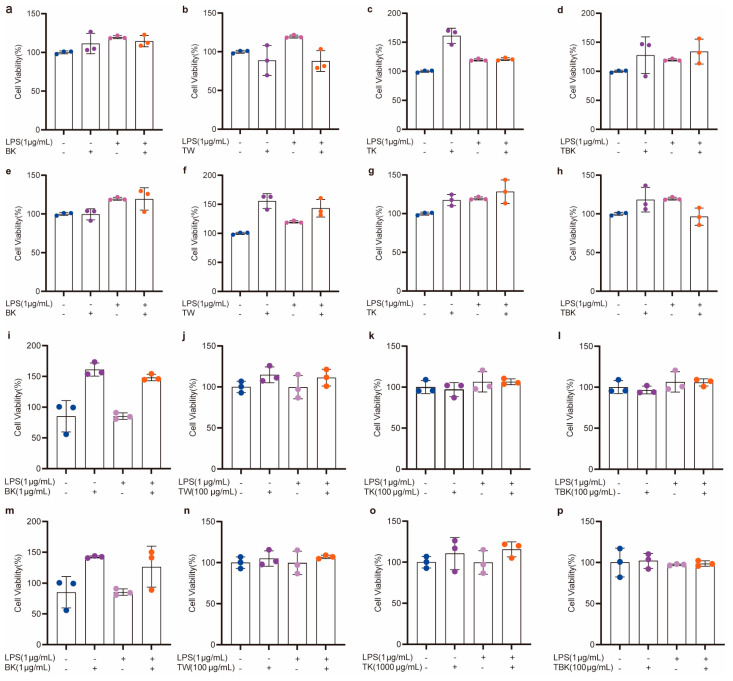
Cytotoxicity of turmeric fermentation broth to RAW264.7 cells stimulated by LPS. (**a**–**d**) Effects of unlyophilized turmeric group stock solution (7 d) on RAW264.7 cells; (**e**–**h**) effects of unlyophilized turmeric group stock solution (14 d) on RAW264.7 cells; (**i**–**l**) effects of lyophilized powder (7 d) in the turmeric group on viability of RAW264.7 cells; (**m**–**p**) effects of lyophilized powder (14 d) in the turmeric group on viability of RAW264.7 cells after LPS treatment.

**Figure 5 ijms-24-13984-f005:**
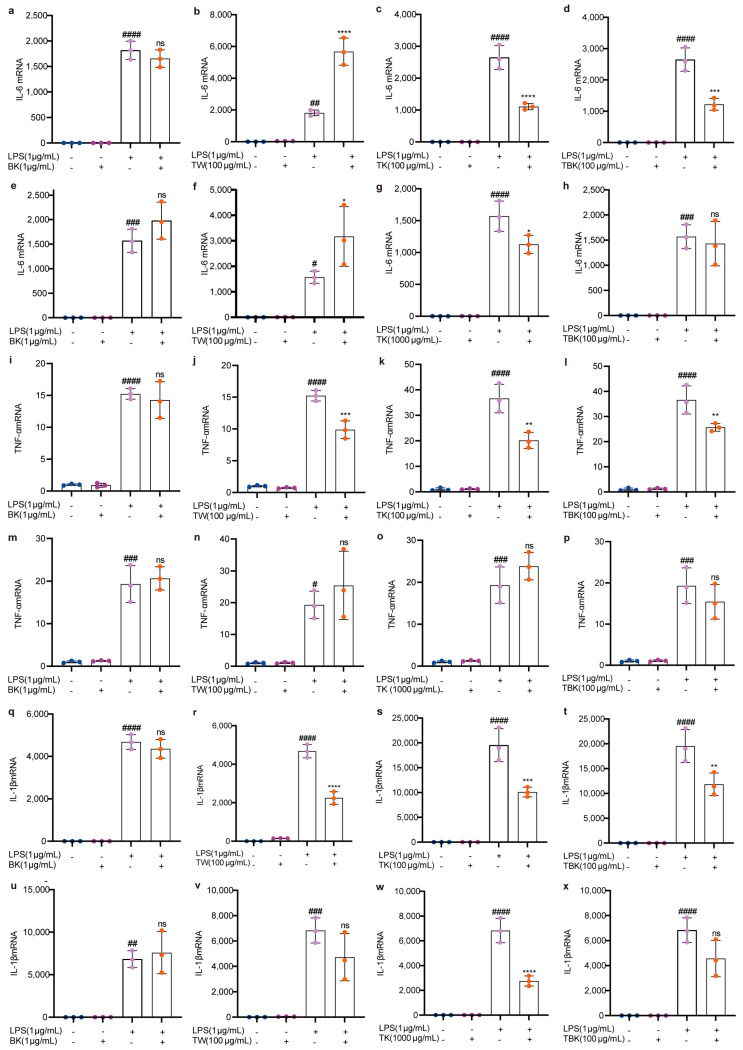
Effects of different samples on IL–6, TNF–α, and IL–1β mRNA expressions in RAW264.7 cells stimulated by LPS. (**a**–**d**) IL–6 mRNA (7 d fermentation); (**e**–**h**) IL–6 mRNA (14 d fermentation); (**i**–**l**) TNF-α mRNA (7 d fermentation); (**m**–**p**) TNF–α mRNA (14 d fermentation); (**q**–**t**) IL–1β mRNA (7 d fermentation); (**u**–**x**) IL–1β mRNA (14 d fermentation). ns: *p* > 0.05; *, #, *p* < 0.05; **, ##, *p* < 0.01; ***, ###, *p* < 0.001; ****, ####, *p* < 0.0001.

**Figure 6 ijms-24-13984-f006:**
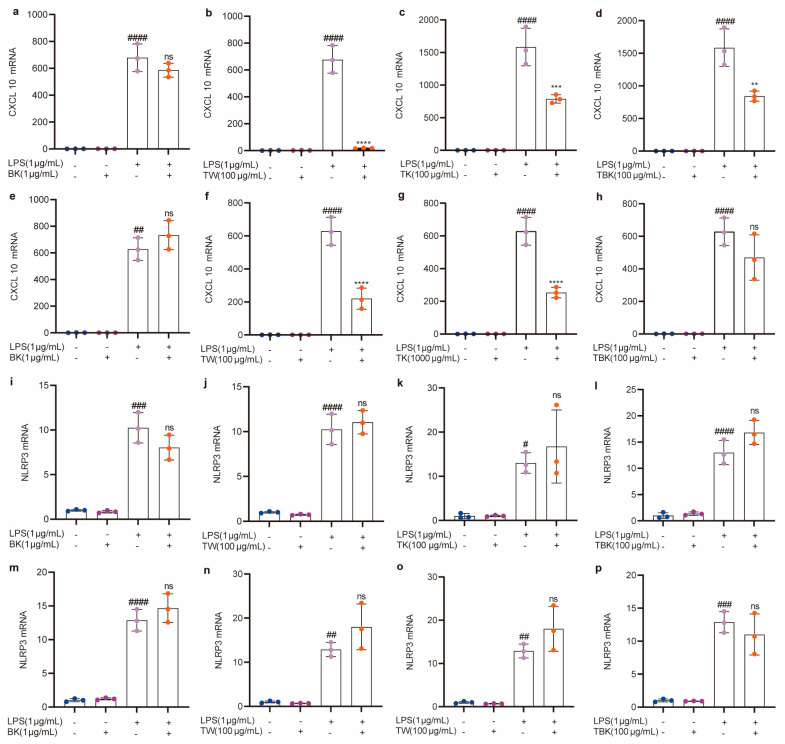
Effects of different samples on CXCL10 and NLRP3 mRNA expressions in RAW264.7 cells stimulated by LPS. (**a**–**d**) CXCL10 mRNA (7 d fermentation); (**e**–**h**) CXCL10 mRNA (14 d fermentation); (**i**–**l**) NLRP3 mRNA (7 d fermentation); (**m**–**p**) NLRP3 mRNA (14 d fermentation). BK, black tea kombucha; TE, turmeric water extract; TK, turmeric kombucha; TBK, turmeric–black tea kombucha. Ns: *p* > 0.05; #, *p* < 0.01; **, ##, *p* < 0.01; ***, ###, *p* < 0.001; ****, ####, *p* < 0.0001.

**Table 1 ijms-24-13984-t001:** Bacteriostatic minimal inhibitory concentration (MIC) of turmeric group samples against *E. coli* (χ ± s, N = 3).

Fermentation Time/d	BK	TE	TK	TBK
7	2.5 ± 0	-	5 ± 0	2.5 ± 0
14	1.25 ± 0	-	2.5 ± 0	2.5 ± 0

BK, black tea kombucha; TE, turmeric water extract; TK, turmeric kombucha; TBK, turmeric–black tea kombucha.

**Table 2 ijms-24-13984-t002:** Bacteriostatic MIC of turmeric samples against *S. aureus* (χ ± s, N = 3).

Fermentation Time/d	BK	TE	TK	TBK
7	1.25 ± 0	-	5 ± 0	1.25 ± 0
14	0.625 ± 0	-	1.25 ± 0	1.25 ± 0

**Table 3 ijms-24-13984-t003:** Bacteriostatic MIC of samples from the *Paeoniae alba* group against *E. coli* (χ ± s, N = 3).

Fermentation Time/d	BK	PE	PK	TPK
7	2.5 ± 0	-	1.25 ± 0	1.25 ± 0
14	0.625 ± 0	2.5 ± 0	0.625 ± 0	0.3125 ± 0

BK, black tea kombucha; PE, *Paeoniae alba* extract; PK, *Paeoniae alba* kombucha; TPK, *Paeoniae alba*–black tea kombucha.

**Table 4 ijms-24-13984-t004:** Antibacterial MIC of samples from the *Paeoniae alba* group against *S. aureus* (χ ± s, N = 3).

Fermentation Time/d	BK	PE	PK	TPK
7	1.25 ± 0	0.3125 ± 0	0.625 ± 0	1.25 ± 0
14	0.3125 ± 0	0.3125 ± 0	0.3125 ± 0	0.3125 ± 0

**Table 5 ijms-24-13984-t005:** Main experimental drugs and reagents.

Name and Specification of Drugs	Production Company
Black tea (Zhengshan race)	Haomingtian Tea Co., Ltd., Wuyishan, China.
Tryptone	Oxoid; Lenexa, KS, USA.
Yeast extract	Guangdong Huankai Biotechnology Co., Ltd., Guangzhou City, China.
NaCl	National Pharmaceutical Group Chemical Reagent Co., Ltd., Shanghai, China.
1,1-Diphenyl-2-picrylhydrazyl radical 2,2-Diphenyl-1-(2,4,6-trinitrophenyl)hydrazyl (DPPH)	Shanghai McLean Biochemical Technology Co., Ltd., Shanghai, China.
vitamin C	Sigma-Aldrich; Saint Louis, MO, USA.
DMEM/HIGH GLUCOSE	Cytiva, Logan, UT, USA.
Penicillin–streptomycin double antibody	Beijing Soleibao Technology Co., Ltd., Beijing, China.
Gibco™ FBS	Samufei Invitrogen, Shanghai, China.
Trypan blue dye	Beijing Soleibao Technology Co., Ltd., Beijing, China.
Trypsin	HyClone Company of USA, Logan, UT, USA.
Lipopolysaccharide (LPS)	Cell Signaling Technology, Danvers, MA, USA.
CCK-8 Kit	Beijing Biyuntian Institute of Biotechnology, Beijing, China.
Trizol	Life Technologies Corporation, Carlsbad, CA, USA.
Trichloromethane	National Pharmaceutical Group Chemical Reagent Co., Ltd., Shanghai, China.
Isopropyl alcohol propan-2-ol	Sigma-Aldrich. St. Louis, MO, USA.
ethanol	Xilong chemical co., Ltd., Chengdu, China.
RNA reverse transcription kit	Mona Biotechnology Co., Ltd., Suzhou, China.
qPCR SYBR Green Master Mix	Yisheng Biotechnology Co., Ltd., Shanghai, China.
Cell Culture Plate	Wuxi NEST Biotechnology Co., Ltd., Wuxi, China.

## Data Availability

All data included in this study are available upon request by contact with the corresponding author.

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
