# Peer review of "Application of Kombucha Fermentation Broth for Antibacterial, Antioxidant, and Anti-Inflammatory Processes"

_ijms, 2023, doi:10.3390/ijms241813984_

Round 1
Reviewer 1 Report
In this manuscript, the authors explore bacteriostatic, anti-oxidant and anti-inflammatory properties of kombucha (turmeric, white peony, alone or added in black tea) fermentation broth as well as respective properties of turmeric and whit peony extracts. They conclude that aforementioned fermentation broths exhibit antibacterial, antioxidant activity and inti-inflammatory. Also, fermentation restricts cytotoxicity of these substrates. It is an interesting study exploring alternative approaches to sepsis management.
Other points:
Please clarify if there is difference between white peony and Radix Paeoniae alba. If there is no difference, afterwards use one term. Note alba, NOT Alba.
Line 40-43: “Owing to antibiotics’ abuse and emergence of multidrug-resistant strains, traditional Chinese medicine has received increasing attention from researchers [8]” instead of “Owing to the abuse of antibiotics, the emergence of multidrug-resistant strains, which make traditional Chinese medicine potentially useful in preventing or treating diferent diseases, has received increasing attention from researchers [8] “
Line 47: Rewrite as “to overexpression of reactive oxygen species”…
Line 64-65: “Turmeric is not only a natural food seasoning and pigment material, but also a traditional Chinese medicine containing bioactive compounds, among which polyphenols…” instead of “Turmeric is not only a traditional Chinese medicine containing bioactive compounds but also a natural food seasoning and pigment material, among which polyphenols..”
Line 82-83: “It is expected that the results of this study will promote kombucha as a potential prevention and treatment for sepsis. “ , as a potential what? Agent?. Please complete the sentence.
Table 3 and 4 : TE TB TBK are correct? Or PE, PK, PBK as far as you refer to peony group.
Line 126-7: please rewrite the following sentence: “the antioxidant capacity of peony culture medium extract was not significantly improved compared with that of peony culture medium ex tract (P>0.05)”
Line 155: “Peony” insread of “peony”
Figure 3: there is inconsistency between the legend and the figure. Please, correct appropriately.
Line 159: Red Camellia paeoniae ? In these experiments, white peony was used, is this right? Did you also study red camelia? Please clarify.
Line 164-5: “stimulated by LPS (Figure 4A-P)” instead of “stimulated by RAW264.7 and LPS (Figure 4A-P)”.
Line 216: “C. longa “? Please clarify if there is difference between turmeric and Curcuma longa. Please do not use different terms.
Line 224 “C. longa “ italics
Line 293: “Upon LPS stimulation of RAW264.7 cells” instead of “After LPS stimulated RAW264.7 cell”
Line 333 and 335: “Clostridium butyricum ” should be written in italics
Line 331: delete “ induced by LPS “
Line367-370: there are black tea fungi (367) or black tea bacteria (369) or both?
Line 390-412: bacteriostatic effect of extracts and fermentation broths. As far as the bacteria (S. aureus and E. coli ) were incubated for 7 and 14 days, I would like to know if new broth was added during this period.
Line 53, 55, 58, 256, 261, 279, etc: “et al. “ in italics
Line 391: please describe plates eg flat/round bottom, TC-treated or not, provider…
Line 424: information is lacking also for 6-well plates
Moderate editing of English language would be required.
Author Response
26 Aug 2023
Prof. Dr. Maurizio Battino
Editor-in-Chief
Ms. Nichapat Rojjananavin
Associate Editor
International Journal of Molecular Sciences
Dear Editor,
I wish to re-submit the manuscript titled “Application of kombucha fermentation broth for antibacterial, antioxidant, and anti-inflammatory processes.” The manuscript ID is ijms-2581228.
We express our gratitude to you and the reviewers for your valuable suggestions and insightful perspectives. The manuscript has greatly benefited from these astute recommendations.
Attached is the revised version of our manuscript. In the following pages are our point-by-point responses to each of the comments of the reviewers. Revisions in the text are highlighted by the utilization of the color red. We hope that the revisions in the manuscript and our accompanying responses would be sufficient to make our manuscript suitable for publication in Frontiers in Microbiology.
Thank you for your consideration. I look forward to hearing from you.
Sincerely,
Jingqian Su, Ph.D.
Associate Professor
Fujian Key Laboratory of Innate Immune Biology
Biomedical Research Center of South China
College of Life Science, Fujian Normal University
Fuzhou 350117, Fujian, China
Tel: +86-18950498937
E-mail: sjq027@fjnu.edu.cn
Minhe Yang, Ph.D.
Professor
College of Life Science, Fujian Normal University,
Fuzhou 350117, China
minhe214@fjnu.edu.cn
Responses to the comments of Reviewer #1
- Please clarify if there is difference between white peony and Radix Paeoniae alba. If there is no difference, afterwards use one term. Note alba, NOT Alba. Line 155: “Peony” insread of “peony”
Response:
We are very grateful for the valuable suggestions from the reviewers. We have unified the terminology of paeoniae alba in the text,as follows:
(1) Line 17
paeoniae alba, and black tea were used as fermenta
(2) Line 21
- aureus and that of paeoniae alba against S. aureus
(3) Line 24
paeoniae alba culture media as substrates
(4) Line 79
turmeric, paeoniae alba, and
(5) Line 93-94
the extract of paeoniae alba culture medium (PW)
(6) Line 96
effect of the paeoniae alba extract
(7) Line 104
paeoniae alba extract had the strongest
(8) Line 114-117
Table 3. Bacteriostatic MIC of samples from the paeoniae alba group against E. coli (c ± s, N = 3). BK: black tea kombucha; PE: paeoniae alba extract; PK: paeoniae alba kombucha; TPK: paeoniae alba; black Tea Kombucha Table 4. Antibacterial MIC of samples from the paeoniae alba group against S. aureus ((c ± s, N = 3).
(9) Line 125-128
lyophilized paeoniae alba group (7 d); (f) Inhibition of E. coli in the paeoniae alba group (14 d); (g) Inhibition of lyophilized powder on S. aureus in the paeoniae alba group (7 d); (h) Inhibition of lyophilized powder on S. aureus in the paeoniae alba group (14 d). PE: paeoniae alba extract; PK: paeoniae alba kombucha; TPK: paeoniae alba -black tea kombucha
(10) Line 157-161
the paeoniae alba group; (g) Scavenging of DPPH radicals by lyophilized powder (14 d) in the paeoniae alba group; (h) 1.25-fold change in DPPH clearance of the paeoniae alba group concentrate (7 d) with time; (i) 1.25-fold change in DPPH clearance of the paeoniae alba group concentrate (14 d) with time. PE: paeoniae alba extract; PK; paeoniae alba kombucha; TPK: paeoniae alba -black tea kombucha.
(11) Line 181-189
paeoniae alba fermentation broth-treated L929 and RAW264.7 cell viabilities following 24 h. (a)Effect of turmeric fermentation broth (7 d) on L929 cell viability; (b) Effect of turmeric fermentation broth (14 d) on L929 cell viability; (c)Effect of paeoniae alba group fermentation broth (7 d) on L929 cell viability; (d) Effect of paeoniae alba group fermentation broth (14 d) on L929 cell viability; (e) Effect of turmeric fermentation broth (7 d) on RAW264.7 cell viability; (f) Effect of turmeric fermentation broth (14 d) on RAW264.7 cell viability; (g) Effect of paeoniae alba fermentation broth (7 d) on RAW264.7 cell viability; (h) Effect of paeoniae alba fermentation broth (14 d) on RAW264.7 cell viability. BK stands for kombucha; TW: turmeric medium; TK is turmeric kombucha; TBK is turmeric kombucha; PW is paeoniae alba culture medium;
(12) Line 241-242
- longa and paeoniae alba.
(13) Line 247-248
the paeoniae alba group (35.0 g/L vs. 30.0 g/L), and the negative value in the turmeric group was larger than that in the paeoniae alba group.
(14) Line 256-257
turmeric and paeoniae alba, probably because the polyphenols in turmeric and paeoniae alba were transformed into substances without DPPH
(15) Line 262-263
paeoniae alba.
(16) Line 269
paeoniae alba.
(17) Line 278-279
fermented paeoniae alba root extract with plant-derived Lactobacillus brevis 174A, and the fermented paeoniae alba root extract
(18) Line 306
paeoniae alba
(19) Line 318-319
paeoniae alba
(20) Line 324
paeoniae alba
(21) Line 358
paeoniae alba
(22) Line 373
paeoniae alba
(23) Line 387-390
paeoniae alba bacteria, and paeoniae alba black tea media for 7 d until the bacteria grew stably, and the mother liquids of turmeric, turmeric black tea, paeoniae alba, and paeoniae alba black tea bacteria
(24) Line 398-400
paeoniae alba group; the preparation method of fermentation broth in the paeoniae alba group is similar, but the difference is: 6 g black tea, 30 g white granulated sugar, 20 g paeoniae alba.
- Line 40-43: “Owing to antibiotics’ abuse and emergence of multidrug-resistant strains, traditional Chinese medicine has received increasing attention from researchers [8]” instead of “Owing to the abuse of antibiotics, the emergence of multidrug-resistant strains, which make traditional Chinese medicine potentially useful in preventing or treating diferent diseases, has received increasing attention from researchers [8]
Response:
Many thanks to the reviewers for their valuable suggestions. According to the suggestions of reviewers, we have revised the sentence in Line 43-44, as follows:
Owing to antibiotics’ abuse and emergence of multidrug-resistant strains, traditional Chinese medicine has received increasing attention from researchers [8].
- Line 47: Rewrite as “to overexpression of reactive oxygen species”…
Response:
We would like to extend our heartfelt appreciation for the invaluable suggestion put forth by the reviewer. The sentence has been rewritten in Lines 47-50, as follow:
In human diseases related to overexpression of reactive oxygen species (ROS) and free radicals, including inflammatory diseases, cancer, senile diabetes, neurodegenerative diseases, and arteriosclerosis, antioxidants and enzymes can be used to scavenge these species [10].
- Line 64-65: “Turmeric is not only a natural food seasoning and pigment material, but also a traditional Chinese medicine containing bioactive compounds, among which polyphenols…” instead of “Turmeric is not only a traditional Chinese medicine containing bioactive compounds but also a natural food seasoning and pigment material, among which polyphenols.
Response:
We express our gratitude for your valuable comment and extend our sincere apologies for any confusion that may have arisen. The sentence has been modified in Lines 65-66, as follow:
Turmeric is not only a natural food seasoning and pigment material, but also a traditional Chinese medicine containing bioactive compounds, among which polyphenols and curcumin derivatives possess antioxidant, anti-inflammatory, and anticancer properties [16].
- Line 82-83: “It is expected that the results of this study will promote kombucha as a potential prevention and treatment for sepsis.”, as a potential what? Agent?. Please complete the sentence.
Response:
We would like to extend our heartfelt appreciation for the invaluable suggestion put forth by the reviewer. The sentence has been corrected in Lines 82-84, as follow:
The results of this study are expected to indicate that kombucha can be used as a potent anti-inflammatory agent for sepsis prevention and treatment.
- Table 3 and 4: TE TB TBK are correct? Or PE, PK, PBK as far as you refer to peony group.
Response:
We express our gratitude to the reviewers for their invaluable comments, which have greatly assisted us. We made the modification in Lines 114-118, as follows:
Table 3. Bacteriostatic MIC of samples from the paeoniae alba group against E. coli (c ± s, N = 3).
|
Fermentation time/d |
BK |
PE |
PK |
TPK |
|
7 |
2.5 ± 0 |
- |
1.25 ± 0 |
1.25 ± 0 |
|
14 |
0.625 ± 0 |
2.5 ± 0 |
0.625 ± 0 |
0.3125 ± 0 |
BK: black tea kombucha; PE: paeoniae alba extract; PK: paeoniae alba kombucha; TPK: paeoniae alba black Tea Kombucha
Table 4. Antibacterial MIC of samples from the paeoniae alba group against S. aureus (c ± s, N = 3).
|
Fermentation time/d |
BK |
PE |
PK |
TPK |
|
7 |
1.25 ± 0 |
0.3125 ± 0 |
0.625 ± 0 |
1.25 ± 0 |
|
14 |
0.3125 ± 0 |
0.3125 ± 0 |
0.3125 ± 0 |
0.3125 ± 0 |
- Line 126-7: please rewrite the following sentence: “the antioxidant capacity of peony culture medium extract was not significantly improved compared with that of peony culture medium ex tract (P>0.05)”
Response:
We would like to extend our heartfelt appreciation for the invaluable suggestion provided by the reviewer. We made the modification in Lines137-138, as follows:
The antioxidant capacity of the PK was not significantly improved compared to that of that PW (P > 0.05)
- Line 155: “Peony” insread of “peony”
- Figure 3: there is inconsistency between the legend and the figure. Please, correct appropriately.
Response:
We express our gratitude for your valuable comment and extend our sincere apologies for any confusion that may have arisen. We used different colors to distinguish different fermentation broths in the same group, in which A-D is turmeric and E-H is paeoniae alba.
- Line 159: Red Camellia paeoniae ? In these experiments, white peony was used, is this right? Did you also study red camelia? Please clarify.
Response:
We would like to extend our heartfelt appreciation for the invaluable suggestion put forth by the reviewer. Here we describe errors and are therefore misleading. The sentence has been corrected in Lines 173-174, as follow:
TPK could significantly promote the proliferation of RAW264.7 cells, but the prolongation of fermentation time reduced the promoting effect.
- Line 164-5: “stimulated by LPS (Figure 4A-P)” instead of “stimulated by RAW264.7 and LPS (Figure 4A-P)”.
Response:
We express our gratitude for your valuable comment and extend our sincere apologies for any confusion that may have arisen. The sentence has been corrected in Lines 176-179, as follow:
The fermentation broth and lyophilized powder from the turmeric group had no inhibito-ry effect on the viability of RAW264.7 cells stimulated by LPS (Figure4a-p).
- Line 216: “C. longa “? Please clarify if there is difference between turmeric and Curcuma longa. Please do not use different terms.
Response:
We would like to extend our heartfelt appreciation for the invaluable suggestion provided by the reviewer. Although both have turmeric meanings, we have replaced with turmeric in our in Line 22, 92, 100, 134, 194, 225, and 361.
- Line 224 “C. longa“ italics.”
Response:
Corrected.
- Line 293: “Upon LPS stimulation of RAW264.7 cells” instead of “After LPS stimulated RAW264.7 cell”
Response:
We would like to extend our heartfelt appreciation for the invaluable suggestion provided by the reviewer. The sentence has been corrected in Lines 334-335, as follows:
Upon LPS stimulation of RAW264.7 cells, the mRNA expressions of inflammatory factors, such as IL-6, IL-1β, TNF-α, and CXCL10, significantly increased (P < 0.05).
- Line 333 and 335: “Clostridium butyricum” should be written in italics
Response:
Corrected.
- Line 331: delete “ induced by LPS”.
Response:
Corrected.
- Line367-370: there are black tea fungi (367) or black tea bacteria (369) or both?
Response:
We express our gratitude for your valuable comment and extend our sincere apologies for any confusion that may have arisen. Two words are used to express the same substance. For clarity, we have chosen “black tea bacteria” in text.
- Line 390-412: bacteriostatic effect of extracts and fermentation broths. As far as the bacteria (S. aureus and E. coli) were incubated for 7 and 14 days, I would like to know if new broth was added during this period.
Response:
Thank you for your query. E. coli and S. aureus were first inoculated on a Luria-Bertani (LB) solid medium plate via plate marking method and cultured in a biochemical incubator (37 ℃) for 24 h to obtain a single colony. A bacterial suspension [1.5 × 106 colony-forming unit (CFU) /mL] was prepared for follow-up experiments.
When made antibacterial activity test (MIC), freeze-dried concentrated fermentation broth and a 1.5 μL concentration of 1.5 × 106 CFU/mL bacterial suspension (A1) was added to each of the first three wells. No additional new broth has been added.
- Line 53, 55, 58, 256, 261, 279, etc: “et al.” in italics
Response:
We would like to extend our heartfelt appreciation for the invaluable suggestion put forth by the reviewer. We have corrected them in the text.
20.Line 391: please describe plates eg flat/round bottom, TC-treated or not, provider…
Response:
We express our gratitude for your valuable comment and extend our sincere apologies for any confusion that may have arisen. We have made additions in Line 385-390, as follows:
4.3. Domestication and culture of black tea fungi in different media preparations of fermentation broth
Black tea bacteria were subcultured in TW, turmeric black tea, paeoniae alba bacteria, and paeoniae alba black tea media for 7 d until the bacteria grew stably, and the mother liquids of turmeric, turmeric black tea, paeoniae alba, and paeoniae alba black tea bacteria were obtained.
- Line 424: information is lacking also for 6-well plates.
Response:
We would like to extend our heartfelt appreciation for the invaluable suggestion provided by the reviewer. The information has been added in Table 5, as follows:
Table 5. Main experimental drugs and reagents.
|
Name and specification of drugs |
Production company |
|
Black tea (Zhengshan race) |
Haomingtian Tea Co., Ltd. |
|
Tryptone |
Oxoid |
|
Yeast extract |
Guangdong Huankai Biotechnology Co., Ltd. |
|
NaCl |
National Pharmaceutical Group Chemical Reagent Co., Ltd. |
|
1,1-Diphenyl-2-picrylhydrazyl radical 2,2-Diphenyl-1-(2,4,6-trinitrophenyl)hydrazyl (DPPH) |
Shanghai McLean Biochemical Technology Co., Ltd. |
|
vitamin C |
Sigma-Aldrich |
|
DMEM/HIGH GLUCOSE |
Cytiva |
|
Penicillin-streptomycin double antibody |
Beijing Soleibao Technology Co., Ltd. |
|
Gibco™ FBS |
Samufei Invitrogen |
|
Trypan blue dye |
Beijing Soleibao Technology Co., Ltd. |
|
Trypsin |
HyClone Company of USA |
|
Lipopolysaccharide (LPS) |
Cell Signaling Technology |
|
CCK-8 Kit |
Beijing Biyuntian Institute of Biotechnology |
|
Trizol |
Life Technologies Corporation |
|
Trichloromethane |
National Pharmaceutical Group Chemical Reagent Co., Ltd. |
|
Isopropyl alcohol propan-2-ol |
Sigma-Aldrich |
|
ethanol |
Xilong chemical co., Ltd. |
|
RNA reverse transcription kit |
Mona Biotechnology Co., Ltd. |
|
qPCR SYBR Green Master Mix |
Yisheng Biotechnology Co., Ltd. |
|
Cell Culture Plate |
Wuxi NEST Biotechnology Co., Ltd. |

Reviewer 2 Report
This is a clever work on current and important topic, however I have some doubts.
Lines 27-28. I don’t like the definition of sepsis, is it based on ref.1?
Line 38 what this phrase mean “enhanced immune activities”?
Section lines 64-74, please add the limitations of this model and underline difficulties in translation of obtained results into the clinical practice. In my opinion the authors statement about the potential for treatment should be justified. There are also no references about potential for prevention of sepsis.
Line 82; only one inflammatory factor was studied ?
Authors did not clearly explain why this work is innovative. Kombucha has been for many years as the food product with many health-promoting properties.
Line 263, explain ROS when first mentioned.
Line 352 were drug resistance or MDR evaluated in this study ?
Please discuss in detail why this model of sepsis research is reliable and useful, and obtained results can be translated into potential clinical applications.
Line 454; I presume this applies to host cells?
Author Response
26 Aug 2023
Prof. Dr. Maurizio Battino
Editor-in-Chief
Ms. Nichapat Rojjananavin
Associate Editor
International Journal of Molecular Sciences
Dear Editor,
I wish to re-submit the manuscript titled “Application of kombucha fermentation broth for antibacterial, antioxidant, and anti-inflammatory processes.” The manuscript ID is ijms-2581228.
We express our gratitude to you and the reviewers for your valuable suggestions and insightful perspectives. The manuscript has greatly benefited from these astute recommendations.
Attached is the revised version of our manuscript. In the following pages are our point-by-point responses to each of the comments of the reviewers. Revisions in the text are highlighted by the utilization of the color red. We hope that the revisions in the manuscript and our accompanying responses would be sufficient to make our manuscript suitable for publication in Frontiers in Microbiology.
Thank you for your consideration. I look forward to hearing from you.
Sincerely,
Jingqian Su, Ph.D.
Associate Professor
Fujian Key Laboratory of Innate Immune Biology
Biomedical Research Center of South China
College of Life Science, Fujian Normal University
Fuzhou 350117, Fujian, China
Tel: +86-18950498937
E-mail: sjq027@fjnu.edu.cn
Minhe Yang, Ph.D.
Professor
College of Life Science, Fujian Normal University,
Fuzhou 350117, China
minhe214@fjnu.edu.cn
Responses to the comments of Reviewer #2
- Lines 27-28. I don’t like the definition of sepsis, is it based on ref.1?
Respond:
We are very grateful for the valuable suggestions from the reviewers. We have revised the sentence in Lines 30-31, as follows:
The latest definition of sepsis is life-threatening organ dysfunction caused by a dysregulated host response to infection [1].
- Line 38 what this phrase mean “enhanced immune activities”?
Respond:
Many thanks to the reviewers for their valuable suggestions. We have corrected the statement in Line 40, as follows:
“and enhanced immunity”
- Section lines 64-74, please add the limitations of this model and underline difficulties in translation of obtained results into the clinical practice. In my opinion the authors statement about the potential for treatment should be justified. There are also no references about potential for prevention of sepsis.
Response:
We would like to extend our heartfelt appreciation for the invaluable suggestion put forth by the reviewer. The inadequacy of LPS-induced models of cellular inflammation has been added in the article, and there is currently no specific prevention method for sepsis. We made the modification in Lines 73-74, as follows:
But can only simulate inflammation, cannot simulate infection in sepsis, etc., so it is not realistic enough.
- Line 82; only one inflammatory factor was studied?
Response:
We express our gratitude for your valuable comment and extend our sincere apologies for any confusion that may have arisen. We have added the inflammatory factors in Line 82, as follows:
“inflammatory factors (IL-6, IL-1β, TNF-α, CXCL10, etc).”
- Authors did not clearly explain why this work is innovative. Kombucha has been for many years as the food product with many health-promoting properties.
Response:
We express our gratitude to the reviewers for their invaluable comments, which have greatly assisted us. Turmeric was employed as a fermentation substrate to produce turmeric kombucha, and subsequent bacteriostatic and anti-inflammatory experiments revealed the favorable antibacterial and anti-inflammatory properties of turmeric kombucha.
- Line 263, explain ROS when first mentioned.
Response:
We would like to extend our heartfelt appreciation for the invaluable suggestion provided by the reviewer. We have made corrections in Line 280, as follows:
“reactive oxygen species (ROS)”
- Line 352 were drug resistance or MDR evaluated in this study?
Response:
We express our gratitude for your valuable comment and extend our sincere apologies for any confusion that may have arisen. We have corrected these in Line 366-367.
- Please discuss in detail why this model of sepsis research is reliable and useful, and obtained results can be translated into potential clinical applications.
Response:
We would like to extend our heartfelt appreciation for the invaluable suggestion put forth by the reviewer. The LPS-induced RAW264.7 cell inflammation model is a widely employed cellular model in sepsis experiments, effectively mimicking the inflammatory response observed in sepsis patients. Our findings indicate that turmeric kombucha exhibits significant inhibitory effects on LPS-induced cellular inflammatory responses, suggesting its potential clinical research value. However, further experimentation is required to validate these observations.
- Line 454; I presume this applies to host cells?
Response:
We express our gratitude for your valuable comment and extend our sincere apologies for any confusion that may have arisen. Based on the results of our toxicity assessments, turmeric kombucha exhibits a notable advantage in terms of low toxicity. Consequently, it is plausible to hypothesize its efficacy against host cells; however, further trials are imperative to validate this assertion.

Round 2
Reviewer 2 Report
I accept the manuscript in present form and recommend it for publication.